# Brain function and metabolism in patients with long-term tacrolimus therapy after kidney transplantation in comparison to patients after liver transplantation

Henning Pflugrad[1,2☯]*, Patrick Nösel[3☯], Xiaoqi Ding[3], Birte Schmitz[3], Heinrich Lanfermann[3], Hannelore Barg-Hock[4], Jürgen Klempnauer[2,4], Mario Schiffer[2,5¤‡], Karin Weissenborn[1,2‡]

1 Department of Neurology, Hannover Medical School, Hannover, Germany, 2 Integrated Research and Treatment Centre Transplantation, Hannover Medical School, Hannover, Germany, 3 Institute of Diagnostic and Interventional Neuroradiology, Hannover Medical School, Hannover, Germany, 4 Clinic for Visceral and Transplant Surgery, Hannover Medical School, Hannover, Germany, 5 Clinic for Nephrology, Hannover Medical School, Hannover, Germany

☯ These authors contributed equally to this work.
¤ Current address: Department of Nephrology and Hypertension, Friedrich-Alexander-Universität Erlangen-Nürnberg, Erlangen, Germany
‡ MS and KW contributed equally and are Joint Senior Authors.
* pflugrad.henning@mh-hannover.de

**Data Availability Statement:** All relevant data are within the paper and its Supporting Information files.

## Abstract

### Background

About 50% of the patients 5–7 years after kidney transplantation show impairment of memory, attention and executive function. Tacrolimus frequently induces neurological complications in the first few weeks after transplantation. Furthermore, tacrolimus treatment is associated with impaired cognitive function in the long-term in patients after liver transplantation. We hypothesize that long-term tacrolimus therapy is associated with cognitive dysfunction and alterations of brain structure and metabolism in patients after kidney transplantation.

### Methods

Twenty-one patients 10 years after kidney transplantation underwent cognitive testing, magnetic resonance imaging and whole brain 31-phosphor magnetic resonance spectroscopy for the assessment of brain function, structure and energy metabolism. Using a cross-sectional study design the results were compared to those of patients 1 (n = 11) and 5 years (n = 10) after kidney transplantation, and healthy controls (n = 17). To further analyze the share of transplantation, tacrolimus therapy and kidney dysfunction on the results patients after liver transplantation (n = 9) were selected as a patient control group.

**Funding:** This study was supported partly by the German Research Foundation (XD) and by a grant from the German Federal Ministry of Education and Research (HP, reference number: 01EO1302). We acknowledge support by the German Research Foundation (DFG) and the Open Access Publication Fund of Hannover Medical School (MHH). The funders had no role in study design, data collection and analysis, decision to publish, or preparation of the manuscript.

**Competing interests:** The authors have declared that no competing interests exist.

**Abbreviations:** ATP, Adenosine triphosphate; CI, confidence interval; CNI, calcineurin inhibitor; FLAIR, T2-weighted-fluid-attenuated inversion recovery sequence; FOV, field of view; GFR, glomerular filtration rate; GRE, T2*-weighted gradient-echo sequence; IQ, interquartile; KT, kidney transplantation; MPRAGE, T1-weighted 3D Magnetization Prepared Rapid Gradient Echo; MRI, magnetic resonance imaging; 31P-MRS, 31-phosphor magnetic resonance spectroscopy; n, number; n.a, not applicable; NAD, Nicotinamide adenine dinucleotide; PCr, Phosphocreatine; PDE, Phosphodiester; Pi, Inorganic phosphate; PME, Phosphomonoester; PVH, periventricular hyperintensities; RBANS, Repeatable Battery for the Assessement of Neuropsychological Status; SD, standard deviation; TE, echo time; TI, Inversion time; TR, repetition time; TSE, T2 weighted turbo spin echo sequence; VS, voxel size; VWCN, Ventricular width at the level of the caudate nucleus; VWSC, Ventricular width at the level of the semioval centre; WMH, white matter hyperintensities.

## Results

Patients 1 and 10 years after kidney transplantation (p = 0.02) similar to patients 10 years after liver transplantation (p<0.01) showed significantly worse cognitive function than healthy controls. In contrast to patients after liver transplantation patients after kidney transplantation showed significantly reduced adenosine triphosphate levels in the brain compared to healthy controls (p≤0.01). Patients 1 and 5 years after kidney transplantation had significantly increased periventricular hyperintensities compared to healthy controls (p<0.05).

## Conclusions

Our data indicate that cognitive impairment in the long-term after liver and kidney transplantation cannot exclusively be explained by CNI neurotoxicity.

## Introduction

Many patients on dialysis awaiting kidney transplantation (KT) suffer from cognitive impairment [1, 2]. Fortunately, an improvement of cognitive function has been observed 1 year after KT in longitudinal assessments; the KT patients even reached the level of healthy controls after transplantation [3–5]. Interestingly, the long-term outcome of cognitive function after KT has only scarcely been analyzed. The few studies available showed impairment of memory, attention and executive function 5–7 years after KT in about 50% of the patients compared to controls [6, 7]. Considering the favorable course of cognitive function within the first year after KT this finding suggests a secondary decrease of cognitive function in the long-term after KT similar to the course of cognition after liver transplantation [8, 9].

One possible mechanism behind long-term cognitive impairment in patients after KT might be calcineurin inhibitor (CNI) therapy. CNIs, currently tacrolimus, are the standard immunosuppressive therapy for patients after KT because they significantly increase long-term survival rates after transplantation [10, 11]. In consequence, however, long-term adverse effects such as renal dysfunction, malignancy and cardiovascular disease gained importance and triggered a discussion about CNI dose reduction strategies [12]. Interestingly, long-term neurological side effects of CNI therapy have hardly been explored although central nervous system toxicity is one of the most important short term side effects of CNIs after transplantation [13]. Long-term CNI therapy could add to the occurrence of cognitive dysfunction after KT by inducing cerebrovascular atherosclerosis and microangiopathy [12], chronic impairment of the cerebral mitochondrial energy metabolism [14] and/or an alteration of the cerebral immune system with consecutive neurodegeneration [15, 16].

We hypothesize that cognitive function, brain structure and metabolism in patients on long-term standard dose tacrolimus therapy 10 years after KT is significantly altered compared to patients 1 year and 5 years after KT as well as healthy controls but similar to the findings in comparable patients 10 years after liver transplantation.

## Patients and methods

### Patients

152 patients registered in the kidney transplantation outpatient clinic database of Hannover Medical School with a history of KT about 10 years ago were screened for eligibility. The

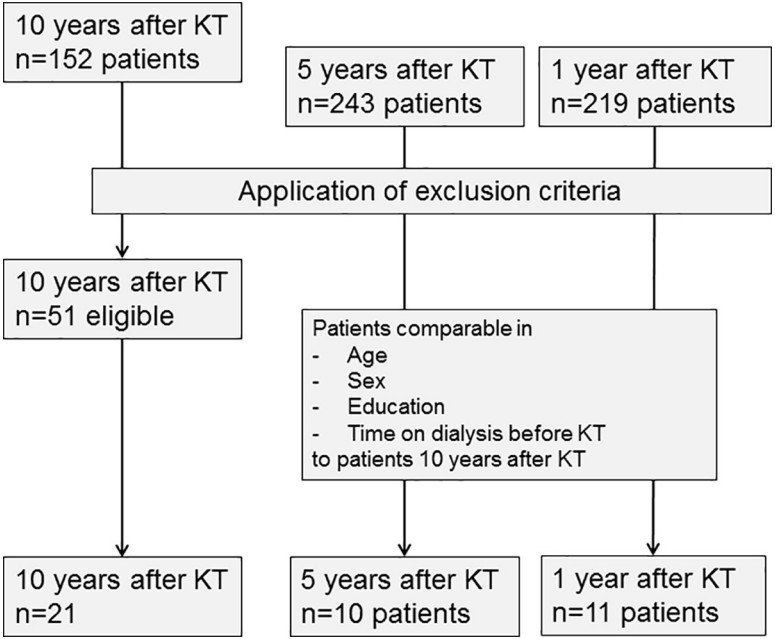

**Fig 1. Flow chart displaying patient selection, application of exclusion criteria and group distribution of patients after kidney transplantation.** This flow chart displays the distribution of patients into the three groups after kidney transplantation.

inclusion criteria were age between 18 and 80 years, German as native language and stable tacrolimus therapy in standard dosage (stable tacrolimus trough levels above 5μg/l). Only patients with tacrolimus therapy were included because it is the standard immunosuppressive drug used after KT [11]. Exclusion criteria were additional transplantation of other organs, kidney re-transplantation (>3 months after first KT), neurological or psychiatric diseases, regular intake of drugs affecting brain function, contraindications for magnetic resonance imaging (MRI), acute transplant-rejection or acute infection and decompensated heart-, liver- or kidney function at study inclusion. After application of the inclusion and exclusion criteria 51 of the patients remained available for study participation. Of these, 21 agreed to participate (Fig 1).

For the cross-sectional analysis it was intended to assess 10 patients 1 and 5 years after KT as patient control groups, respectively. Thus, all patients registered in the kidney transplantation outpatient clinic database about 1 or 5 years after KT were screened to find comparable subjects to the 21 patients 10 years after KT concerning age, sex, education and time on dialysis before KT (Fig 1).

Finally, 42 patients after KT were included: 21 patients about 10 years (KT10), ten patients about 5 years (KT5) and eleven patients about 1 year after KT (KT1) (Table 1). All KT patients were treated with standard dose tacrolimus. However, all patients received at least one further drug for immunosuppression: 32 patients were additionally treated with prednisolone and mycophenolic acid, seven additionally with prednisolone and three additionally with mycophenolic acid. Eight patients of KT10 had been treated with ciclosporin for 261.4±490.4 days before the immunosuppression was changed to tacrolimus.

Nine patients about 10 years after liver transplantation (LT) were selected as a further patient control group. These patients are a subset of 85 patients who participated in a study analysing cognitive function and brain alterations in patients after liver transplantation [8].

**Table 1. Characteristics of the 3 kidney transplantation patient groups, liver transplantation patients and healthy controls.**

| | KT10 | KT5 | KT1 | LT | HC | p value |
|---|---|---|---|---|---|---|
| | **n = 21** | **n = 10** | **n = 11** | **n = 9** | **n = 17** | |
| Age years mean ± SD | 55.9±10.3 | 56.7±6.5 | 54.4±4.5 | 50.3±11.4 | 56.8±8.2 | 0.43 |
| Sex (male/female) | 11/10 | 5/5 | 8/3 | 6/3 | 8/9 | 0.65 |
| Education in years mean ± SD | 10.5±1.8 | 10.6±2.3 | 10.5±1.4 | 10.2±1.9 | 11.4±1.8 | 0.52 |
| Aetiology of kidney disease (n) | | | | n.a. | n.a. | 0.37* |
| Polycystic kidney disease | n = 8 | n = 4 | n = 2 | | | |
| Nephropathy | n = 4 | n = 3 | n = 5 | | | |
| Nephritis | n = 5 | n = 3 | n = 1 | | | |
| other | n = 4 | n = 0 | n = 3 | | | |
| Time on dialysis before KT years mean ± SD | 4.4±3.2 | 3.0±3.4 | 6.3±4.5 | n.a. | n.a. | 0.12* |
| Years since transplantation mean ± SD | 10.8±1.1 | 5.7±0.7 | 1.6±0.7 | 9.7±1.9 | n.a. | **<0.001*** |
| | | | | | | KT10vsLT = 0.50 |
| Donor living/deceased | 7/14 | 8/2 | 3/8 | 0/9 | n.a. | **0.02*** |
| Tacrolimus trough level (µg/l) mean ± SD | 6.9±0.6 | 6.9±0.9 | 8.3±6.8 | 6.8±0.6 | n.a. | **KT10vsKT1 <0.001** |
| | | | | | | **KT5vsKT1 <0.001** |
| | | | | | | **KT1vsLT <0.001** |
| Tacrolimus total dose (g) mean ± SD | 14.3±6.1 | 13.1±6.4 | 2.0±0.6 | 24.7±5.5 | n.a. | **KT10vsKT1 <0.001** |
| | | | | | | **KT10vsLT <0.01** |
| | | | | | | **KT5vsKT1 <0.01** |
| | | | | | | **KT5vsLT <0.01** |
| | | | | | | **KT1vsLT <0.001** |
| Arterial hypertension (+/-) | 21/0 | 10/0 | 11/0 | 5/4 | n.a. | n.a.* |
| Diabetes mellitus (+/-) | 2/19 | 1/9 | 1/10 | 1/8 | n.a. | 0.99* |
| Hypercholesterolemia (+/-) | 15/6 | 4/6 | 6/5 | 1/8 | n.a. | 0.23* |
| GFR mean ± SD | 46.3±16.4 | 47.3±10.3 | 49.5±20.9 | 94.3±15.8 | n.a. | 0.87* |
| Chronic kidney disease grade III (+/-) | 18/3 | 9/1 | 9/2 | 0/9 | n.a. | 0.87* |

KT, kidney transplantation; LT, liver transplantation; HC, healthy control; n, number; SD, standard deviation; n.a., not applicable; others: Goodpasture syndrome n = 1, Membranoproliferative glomerulonephritis n = 1, nephrosclerosis n = 2, kidney shrinkage n = 1, unknown reason n = 2; GFR, glomerular filtration rate in ml/min; +, yes; -, no; p value ≤0.05 is considered significant

*, between KT groups

The subgroup of patients after liver transplantation was selected according to time since transplantation, treatment with standard dose tacrolimus and stable tacrolimus trough levels above 5µg/l, age, education and sex to be comparable to the KT patient cohort. The underlying liver disease of the nine selected patients was primary sclerosing cholangitis (n = 5), polycystic liver disease (n = 2) and hepatitis B virus infection induced cirrhosis (n = 2). Seven liver transplantation patients were treated additionally with at least one further immunosuppressant: three with prednisolone, two with prednisolone and azathioprine, one with prednisolone and mycophenolic acid and one with mycophenolic acid. Five of these 9 patients had been treated with ciclosporin for 91.4±137.8 days before the immunosuppression was changed to tacrolimus.

Data of 33 healthy controls was already available. Of these, 17 (HC) adjusted for age, gender and education were selected and served as a control group (Table 1).

2 KT patients (n = 1 of KT10 and KT1, respectively) had incomplete MRI measurement and thus were excluded from MRI analysis. All subjects gave written informed consent. The study was approved by the local ethics committee at Hannover Medical School and performed according to the World Medical Association Declaration of Helsinki of 1975 (as revised in

2013). No donor organs were obtained from executed prisoners or otherwise institutionalized persons. None of the transplant donors were from a vulnerable population and all donors or next of kin provided written informed consent that was freely given.

## Methods

All subjects underwent a standardised physical neurological examination (H.P.). Age, sex, years of education, underlying kidney disease (diseases were grouped according to polycystic kidney disease, nephropathy, nephritis and other), presence of arterial hypertension, diabetes mellitus or hypercholesterolemia, glomerular filtration rate (GFR) at the time of study inclusion in ml/min, bilirubin total in μmol/l at the time of study inclusion, medication, years since transplantation, living or deceased donor, time on dialysis before KT, tacrolimus dosages and tacrolimus trough levels of each visit at the outpatient clinic were assessed and documented from case records. The GFR was used to classify patients into patients with or without chronic kidney disease grade III. The mean tacrolimus trough level and the total tacrolimus dosage for each patient were calculated with last observations carried forward as previously described [8].

### Psychometric testing and Beck's Depression Inventory

Cognitive function was assessed by the Repeatable Battery for the Assessment of Neuropsychological Status (RBANS) [17–19]. Indications of depression were determined by the Beck's Depression Inventory [20].

### Magnetic resonance imaging and–spectroscopy

All subjects underwent Magnetic resonance imaging and–spectroscopy examinations at 3T (Verio, Siemens, Erlangen, Germany). Cerebrovascular lesions and alterations of brain volume were assessed by semi quantitative assessment with a 12-channel phased-array head coil. The MRI protocol consisted of a T2 weighted turbo spin echo sequence (TSE) with triple echos (repetition time (TR)/echo time (TE) = 6440/8.7/70/131, flip angle 150˚), a T2*-weighted gradient-echo sequence (GRE) with triple echos (TR/TE = 1410/6.42/18.42/30.42 ms, flip angle 20˚), a T1-weighted 3D Magnetization Prepared Rapid Gradient Echo (MPRAGE) acquisition (TR/TE/Inversion time (TI) = 1900/2.93/900 ms, flip angle 9˚), and a T2-weighted-fluid-attenuated inversion recovery sequence (FLAIR) (TR/TE/TI = 9000/94/2500 ms, flip angle 150˚). All scans were made in axial section with an acceleration factor of 2. A field of view (FOV) with 256x208 $mm^2$ and a voxel size (VS) with 1x1x3 $mm^3$ were applied for TSE and GRE scans, while a FOV with 256x224 $mm^2$ and a VS with 1x1x1 $mm^3$ for MPRAGE scan, and a FOV with 230x194 $mm^2$ and a VS with 1x0.9x5 $mm^3$ for FLAIR scan were used. The details of the sequences are displayed in S1 Table. Periventricular hyperintensities (PVH) and white matter hyperintensities (WMH) were assessed visually using the FLAIR and GRE sequences and semi quantitative graded by the Scheltens Scale [21]. Furthermore, the ventricular width at the level of the caudate nucleus (VWCN) and semioval centre (VWSC) were measured in mm using the FLAIR sequence. The assessed structural MRI values are displayed in Fig 2.

For 31-phosphor magnetic resonance spectroscopy (31P-MRS) a non-localized single pulse 31P-MRS free induction decay sequence (TR/TE = 2000/0.23 ms, 64 acquisitions, flip angle of 50˚) was used. A double-tuned 1H/31P transmit/receiver volume head coil (Rapid Biomedical, Würzburg, Germany) was used for 31P-MRS scans. The data of 31P-MR spectra were processed with an adapted LCModel version, with the spectral basis sets calculated by the VeSPA program [22] to estimate global concentrations of the brain phosphorous metabolites Adenosine triphosphate (ATP), Phosphocreatine (PCr), Inorganic phosphate (Pi), Nicotinamide adenine dinucleotide (NAD), Phosphomonoester (PME) and Phosphodiester (PDE). The

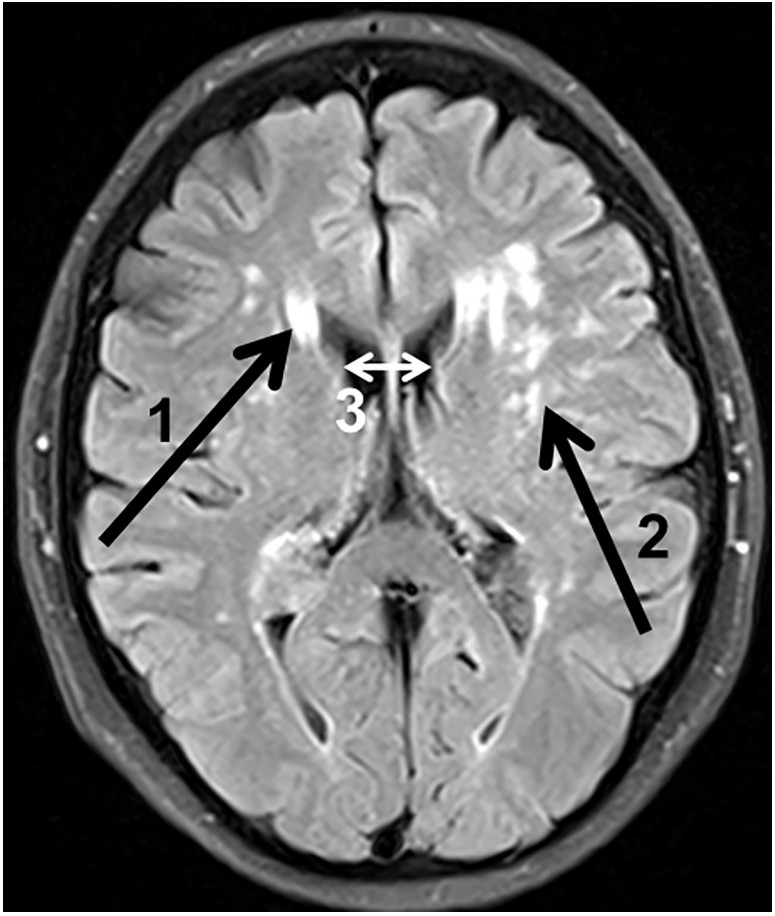

**Fig 2. Exemplary illustration of magnetic resonance imaging parameters.** This exemplary FLAIR image displays the assessed MRI values periventricular hyperintensities (PVH, arrow 1), white matter hyperintensities (WMH, arrow 2) and ventricular widths at the level of the caudate nucleus (VWCN, arrow 3) in a 54 year old female patient 11 years after kidney transplantation.

obtained 31P metabolite values were subsequently corrected for T1 saturation as previously described [23]. An exemplary spectrum of 31P-MRS of a patient (male, 57 years old) together with overlaid LCModel analysis is shown in Fig 3. Each 31P metabolite concentration was determined as a ratio to the sum of total phosphorus metabolite values without unit. The Cramer-Rao lower bound (CRLB) of the spectral analysis was used as quality criterion for estimated metabolite values, as recommended by the author of LCModel [24], i.e. only metabolites estimated with a CRLB less than 25% were considered for further analyses. All MRI and MRS were visually inspected by two experienced neuroradiologists to exclude subjects with morphological abnormalities or artefacts.

## Statistical methods

Kolmogorov-Smirnov test was applied to assess normality of distribution. Kruskal-Wallis-Test and Mann-Whitney-U test for abnormally distributed values and Analysis of variance (ANOVA) with post-hoc Bonferroni correction for normally distributed values were applied to test for significant group differences. Categorical variables were assessed by Chi-squared test. Pearson test (normal distribution) and Spearmen rank test (abnormal distribution) were

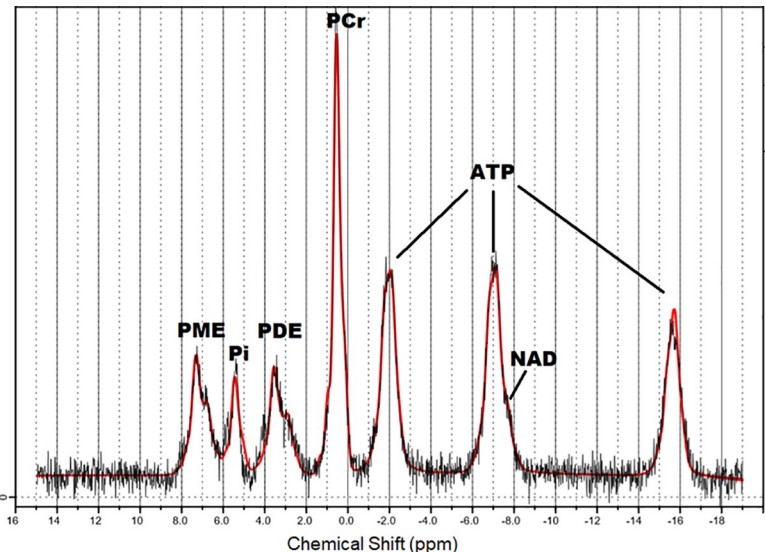

**Fig 3. Exemplary 31P MR spectrum with overlaid LCModel analysis (red line) of a patient (male, 57 years old).**
ATP = adenosine-5-triphosphate; PCr = phosphocreatine; PME = phosphomonoesters; PDE = phosphodiesters;
Pi = inorganic phosphate; NAD = Nicotinamide adenine dinucleotide.

used for correlation analysis. A backward multivariate linear regression analysis was applied to identify independent prognostic factors for cognitive function considering the RBANS Total scale as dependent variable. Independent variables for the analysis including all subjects were age, diagnosis (KT, liver transplantation or healthy control), PVH occipital, ATP and PME. The regression coefficient, p value and confidence interval (CI) are displayed.

Normally distributed values are shown as mean ± standard deviation, abnormally distributed values are shown as median with interquartile (IQ) range. A p-value ≤0.05 was considered significant for all tests applied. The statistical calculations were performed with SPSS 24 (IBM, Armonk, NY).

# Results

## Characteristics of patients and controls

The physical neurological examination was normal in all subjects. Age, sex and years of education did not significantly differ between the patient groups and between patients and healthy controls. In accordance with the design of the study, the time since transplantation differed significantly between the patient groups. The KT1 patients had a significantly higher tacrolimus mean trough level than all other patient groups. Directly after transplantation higher doses of tacrolimus are needed leading to high trough levels. With time after transplantation the tacrolimus dose is reduced. Thus, the time since transplantation explains the significantly higher mean tacrolimus trough level in the KT1 patient group. The time on dialysis before KT, underlying kidney disease, the presence of arterial hypertension, diabetes mellitus, hypercholesterolemia or chronic kidney disease grade 3 according to the GFR did not differ between the KT patient groups. Interestingly, more patients in KT10 and KT1 had received their organ from a deceased donor compared to KT5 (p = 0.02). All patients after liver transplantation had received organs from deceased donors. Compared to the KT patients arterial hypertension (p<0.001) and hypercholesterolemia (p = 0.02) were less often present in liver transplantation patients while diabetes mellitus was equally frequent (p = 0.99). Liver transplantation patients

had a significantly higher GFR at the time of study inclusion and in contrast to all three KT patient groups none of the liver transplantation patients had a chronic kidney disease grade 3 (p<0.001) (Table 1).

## RBANS and Beck's Depression Inventory

All patient groups and controls had a median Beck's Depression Inventory score within the normal range (mean≤6) and the groups did not differ significantly (p = 0.76).

Three (14%) patients of KT10 as well as one patient of KT1 (9%) and LT (11%), respectively, had a pathological RBANS test result (percentile <10%). The mean RBANS results of the index scores immediate memory (p = 0.02), visuospatial/constructional (p = 0.001) and delayed memory (p = 0.02) as well as the RBANS Total scale (p = 0.001) differed significantly between the groups. The pairwise group analysis showed that patients 1 year after KT achieved significantly worse mean results in the index scores immediate memory (91.0±16.2 vs 109.1 ±12.7; p = 0.04) and visuospatial/constructional ability (92.6±14.1 vs 110.7±16.1; p = 0.01) compared to healthy controls. KT patients about 5 and 10 years after transplantation achieved significantly worse mean scores only in visuospatial/constructional ability compared to controls (94.7±13.4 vs 110.7±16.1; p<0.05 and 91.3±11.3 vs 110.7±16.1; p = 0.001, respectively). Concerning the index score delayed memory the pairwise analysis between KT patients about 10 years after transplantation and healthy controls was not significant (p = 0.06). The patients 10 years after liver transplantation scored significantly worse in the RBANS index score immediate memory (89.7±13.7 vs 109.1±12.7; p = 0.04) than healthy controls. The liver transplantation patients showed the worst attention compared to all other groups, however, this was not significant in group comparison (p = 0.08). The patients 1 and 10 years after KT (94.5±13.5 vs 109.4±10.1 and 96.8±12.5 vs 109.4±10.1; both p = 0.02) as well as the liver transplantation patient control group (90.7±9.1 vs 109.4±10.1; p<0.01) achieved a significantly worse RBANS Total scale as a measure for the overall cognitive function compared to healthy controls. The RBANS results among the KT patient groups or between the KT groups and the liver transplantation patient control group did not differ significantly (Table 2 and Fig 4).

**Table 2. RBANS and Beck's Depression Inventory results.**

| | KT10 | KT5 | KT1 | LT | HC | p value |
|---|---|---|---|---|---|---|
| | n = 21 | n = 10 | n = 11 | n = 9 | n = 17 | |
| | mean ± SD | mean ± SD | mean ± SD | mean ± SD | mean ± SD | |
| Beck's Depression Inventory | 6.3±6.8 | 4.0±3.9 | 4.4±4.5 | 5.9±4.6 | 6.0±5.2 | 0.76 |
| Immediate memory | 96.9±19.8 | 98.2±9.5 | 91.0±16.2 | 89.7±13.7 | 109.1±12.7 | **KT1vsHC = 0.04** |
| | | | | | | **LTvsHC = 0.04** |
| Visuospatial/Constructional | 91.3±11.3 | 94.7±13.4 | 92.6±14.1 | 94.9±14.1 | 110.7±16.1 | **KT10vsHC = 0.001** |
| | | | | | | **KT5vsHC <0.05** |
| | | | | | | **KT1vsHC = 0.01** |
| Language | 101.9±10.5 | 106.0±15.4 | 104.6±10.5 | 99.0±12.2 | 105.8±9.1 | 0.57 |
| Attention | 100.7±19.9 | 101.6±12.0 | 94.9±20.5 | 85.1±14.0 | 102.5±8.0 | 0.08 |
| Delayed memory | 98.0±9.6 | 103.8±13.2 | 97.3±11.6 | 97.9±5.0 | 106.1±11.3 | KT10vsHC = 0.06 |
| Total scale | 96.8±12.5 | 100.9±12.8 | 94.5±13.5 | 90.7±9.1 | 109.4±10.1 | **KT10vsHC = 0.02** |
| | | | | | | **KT1vsHC = 0.02** |
| | | | | | | **LTvsHC <0.01** |

KT, kidney transplantation; LT, liver transplantation; HC, healthy control; n, number; SD, standard deviation; p value ≤0.05 is considered significant

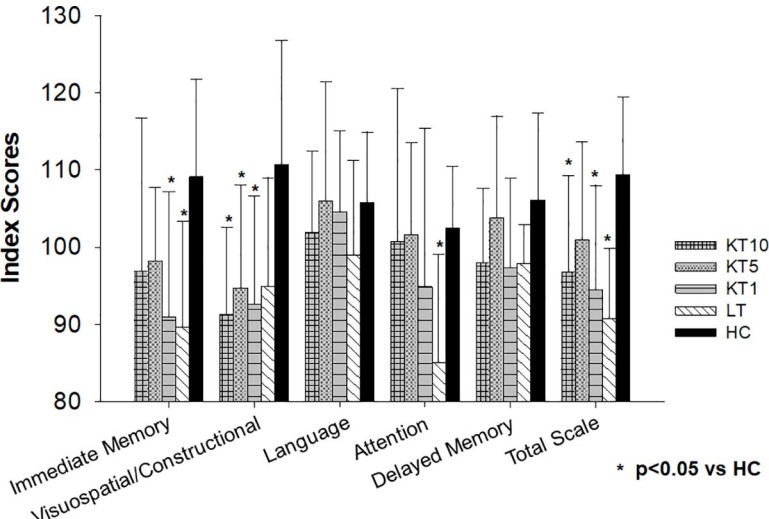

**Fig 4. RBANS results.** This figure displays the RBANS results of the three kidney transplantation groups, the patients after liver transplantation and healthy controls. Each RBANS index score and the Total scale of the groups were compared by Analysis of variance with post-hoc Bonferroni correction for pairwise comparison. Level of significance p<0.05. KT10, 10 years after kidney transplantation; KT5, 5 years after kidney transplantation; KT1, 1 year after kidney transplantation; LT, 10 years after liver transplantation; HC, healthy control.

In conclusion, all patients–after KT and liver transplantation—showed an impaired visuo-spatial/constructional ability compared to healthy controls. Interestingly, only the patients after liver transplantation showed impaired immediate memory and reduced attention compared to healthy controls.

The RBANS results of the KT patients did not correlate with the tacrolimus total dose (S1 Fig) or mean trough level, GFR (S2 Fig), Bilirubin (S3 Fig) or years on dialysis before KT. The variables underlying kidney disease, living or deceased donor, diabetes mellitus or hypercholesterolemia had no effect on the RBANS results.

## Magnetic resonance imaging and spectroscopy

Although data showed higher periventricular hyperintensities and white matter hyperintensities values in general in the patients irrespective of the time interval since transplantation a significant difference from controls could be shown only for a few locations: patients 1 year after KT had a significantly increased extent of occipital and total periventricular hyperintensities compared to healthy controls (p<0.05 and p<0.01, respectively). Furthermore, the patients 5 years after KT showed a significantly increased extent of total periventricular hyperintensities compared to healthy controls (p = 0.01). Interestingly, the KT patients 10 years after transplantation showed no significant differences compared to healthy controls in all assessed MRI parameters. Concerning white matter hyperintensities and the ventricular widths no significant group differences were found (Table 3 and Fig 5A–5C).

The phosphor spectroscopy showed significant differences between the groups for ATP (p<0.001). ATP concentrations in all 3 KT patient groups were significantly reduced compared to healthy controls (10 years after KT p<0.001, 5 years after KT p = 0.001, 1 year after KT p = 0.01) (Table 4 and Fig 6). Concerning the metabolites PCr, Pi, NAD, PDE and PME no significant differences were found between the patient groups and healthy controls. The correlation analysis including only patients after KT, however, showed a significant positive correlation between PME levels and the RBANS domain score Immediate memory (r = 0.43, p<0.01)

**Table 3. Magnetic resonance imaging results.**

|  | KT10 | KT5 | KT1 | LT | HC | p value |
|---|---|---|---|---|---|---|
|  | n = 20 | n = 10 | n = 10 | n = 9 | n = 17 |  |
|  | mean ± SD | mean ± SD | mean ± SD | mean ± SD | mean ± SD |  |
| PVH occipital | 1.4±0.6 | 1.7±0.7 | 1.8±0.4 | 1.6±0.7 | 1.0±0.7 | KT1vsHC <0.05 |
| PVH frontal | 1.4±0.5 | 1.4±0.5 | 1.5±0.7 | 1.1±0.6 | 1.0±0.4 | 0.09 |
| PVH lateral | 0.9±0.6 | 1.1±0.6 | 1.0±0.5 | 0.8±0.4 | 0.6±0.5 | 0.11 |
| PVH total | 3.6±1.1 | 4.2±0.9 | 4.3±1.3 | 3.4±1.4 | 2.7±1.0 | KT5vsHC = 0.01 |
|  |  |  |  |  |  | KT1vsHC <0.01 |
| WMH frontal | 2.8±1.9 | 2.2±1.6 | 2.1±1.7 | 1.8±1.5 | 2.3±1.7 | 0.67 |
| WMH parietal | 1.8±1.8 | 1.9±2.0 | 1.7±2.0 | 1.1±1.5 | 0.6±1.2 | 0.23 |
| WMH occipital | 1.1±1.4 | 0.5±0.9 | 0.6±1.6 | 0.9±1.8 | 0.5±1.4 | 0.76 |
| WMH temporal | 0.6±1.1 | 0.2±0.4 | 0.0±0.0 | 0.3±1.0 | 0.1±0.3 | 0.25 |
| WMH total | 6.1±5.0 | 4.8±3.6 | 4.4±3.1 | 4.1±4.2 | 3.5±3.9 | 0.44 |
| VWCN (mm) | 14.4±3.4 | 14.5±3.6 | 12.9±2.4 | 14.8±2.7 | 13.4±2.9 | 0.54 |
| VWSC (mm) | 29.1±5.8 | 28.8±7.0 | 26.5±5.2 | 31.3±4.1 | 29.4±4.8 | 0.43 |

KT, kidney transplantation; LT, liver transplantation; HC, healthy control; PVH, periventricular hyperintensities; WMH, white matter hyperintensities; VWCN, ventricular width at the level of the caudate nucleus; VWSC, ventricular width at the level of the semioval centre; n, number; SD, standard deviation; p value ≤0.05 is considered significant

and the RBANS Total Scale (r = 0.45, p<0.01). No correlations in patients after KT between ATP (S4 Fig), PCr, Pi, NAD or PDE and ventricular widths, the other RBANS test results, tacrolimus mean trough level or total dose, years on dialysis before KT or GFR were found. The MRI values periventricular hyperintensities (S5 Fig) and white matter hyperintensities showed no association with RBANS test results. Presence or absence of chronic kidney disease

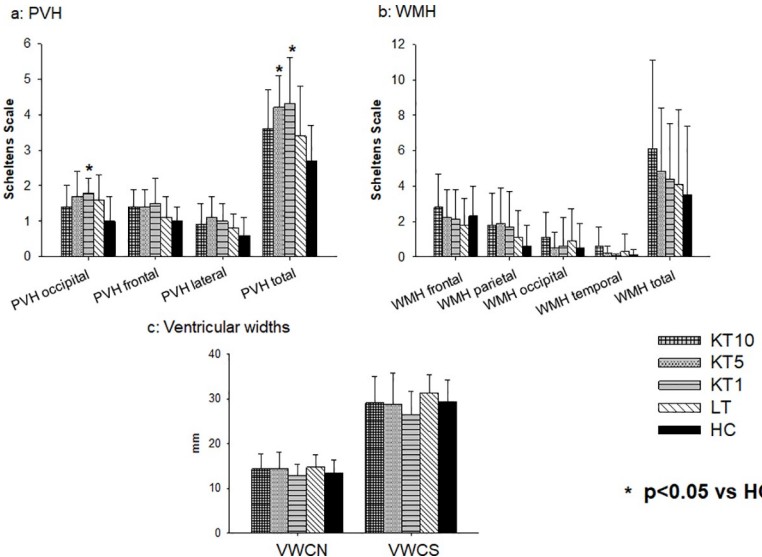

**Fig 5.** a-c: Magnetic resonance imaging results. This figure displays the magnetic resonance imaging results of the three kidney transplantation groups, the patients after liver transplantation and healthy controls. Level of significance p<0.05. KT10, 10 years after kidney transplantation; KT5, 5 years after kidney transplantation; KT1, 1 year after kidney transplantation; LT, 10 years after liver transplantation; HC, healthy control; PVH, periventricular hyperintensities; WMH, white matter hyperintensities; VWCN, ventricular width at the level of the caudate nucleus; VWSC, ventricular width at the level of the semioval centre.

**Table 4. Magnetic resonance spectroscopy results.**

| | KT10 | KT5 | KT1 | LT | HC | p value |
|---|---|---|---|---|---|---|
| | n = 20 | n = 10 | n = 10 | n = 9 | n = 16 | |
| | mean ± SD | mean ± SD | mean ± SD | mean ± SD | mean ± SD | |
| ATP | 0.207±0.009 | 0.206±0.008 | 0.209±0.008 | 0.223±0.020 | 0.224±0.011 | **KT10vsHC <0.001** |
| | | | | | | **KT5vsHC = 0.001** |
| | | | | | | **KT1vsHC = 0.01** |
| PCr | 0.333±0.020 | 0.338±0.023 | 0.330±0.026 | 0.328±0.020 | 0.319±0.018 | 0.23 |
| Pi | 0.079±0.007 | 0.075±0.006 | 0.080±0.007 | 0.083±0.009 | 0.081±0.007 | 0.19 |
| NAD | 0.040±0.005 | 0.042±0.005 | 0.037±0.008 | 0.037±0.006 (n = 8) | 0.039±0.005 | 0.26 |
| PME | 0.212±0.012 | 0.209±0.016 | 0.211±0.012 | 0.218±0.011 | 0.223±0.017 | 0.06 |
| PDE | 0.135±0.019 | 0.133±0.014 | 0.136±0.019 | 0.124±0.018 | 0.127±0.015 | 0.39 |

KT, kidney transplantation; LT, liver transplantation; HC, healthy control; ATP, Adenosine triphosphate; PCr, Phosphocreatine; Pi, Inorganic phosphate; NAD, Nicotinamide adenine dinucleotide; PME, Phosphomonoester; PDE, Phosphodiester; n, number; SD, standard deviation; p value ≤0.05 is considered significant

grade III, diabetes mellitus and hypercholesterolemia had no effect on the MRI/MRS parameters of the KT patient groups.

### Linear regression

A backwards linear regression analysis with the RBANS Total scale as dependent and age, diagnosis (KT, liver transplantation or healthy control), PVH occipital, ATP and PME as independent factors was applied. PME (n = 65, regression coefficient 356.1, p<0.001; 95% CI 166.4–545.8) and being a healthy control or KT patient (n = 65, regression coefficient 8.7, p<0.001; 95% CI 4.3–13.2) were identified as significant positive predictors of the RBANS Total scale. Two KT patients and one healthy control were excluded due to missing MRI.

### Discussion

This single center cross-sectional observational study investigated cognitive function, brain structure and metabolism in patients with standard dose tacrolimus therapy 10 years after

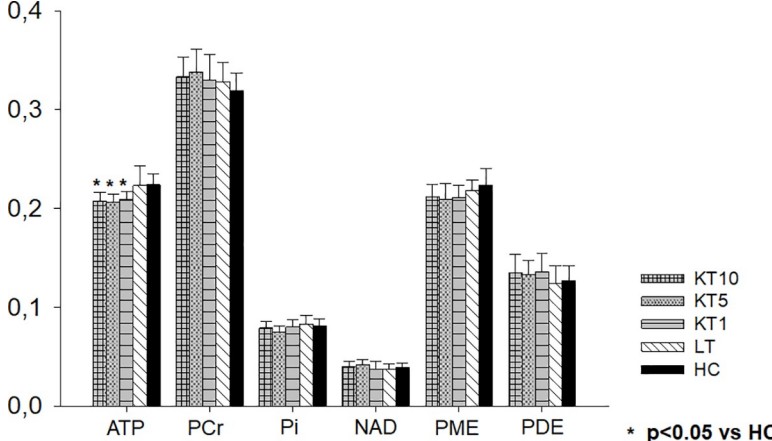

**Fig 6. Magnetic resonance spectroscopy results.** This figure displays the magnetic resonance spectroscopy results of the three kidney transplantation groups, the patients after liver transplantation and healthy controls. Level of significance p<0.05. KT10, 10 years after kidney transplantation; KT5, 5 years after kidney transplantation; KT1, 1 year after kidney transplantation; LT, 10 years after liver transplantation; HC, healthy control; ATP, Adenosine triphosphate; PCr, Phosphocreatine; Pi, Inorganic phosphate; NAD, Nicotinamide adenine dinucleotide; PME, Phosphomonoester; PDE, Phosphodiester.

kidney transplantation. The results were compared to patient control groups consisting of patients 1 and 5 years after KT and tacrolimus treated patients who underwent liver transplantation about 10 years ago as well as healthy controls. Both, patients after KT and patients after liver transplantation showed cognitive impairment compared to healthy controls adjusted for age and education. Furthermore, KT patients showed an increased extent of periventricular hyperintensities and reduced brain ATP concentrations compared to healthy controls.

Tacrolimus is currently the standard immunosuppressive drug used after kidney transplantation to prevent rejection [11]. However, long-term tacrolimus therapy is accompanied by several significant adverse effects such as renal dysfunction, cardiovascular disease and malignancy [12]. Furthermore, tacrolimus therapy is associated with neurological complications in the first few weeks after transplantation [13] and these are expected in the long-term as well. Several possible pathomechanisms are discussed: Tacrolimus induced cardiovascular risk factors [12] enhance atherosclerosis and microangiopathy which may lead to cerebrovascular events and subsequently impaired brain function, tacrolimus might cause neurodegeneration by inhibiting the cerebral immune system [15] and tacrolimus might impair the cerebral energy metabolism [14].

While the improvement of cognitive dysfunction associated with severe chronic kidney disease and hemodialysis within 1 year after KT is well described [3, 4], interestingly, only few studies investigated the long-term neurological outcome after KT. Troen and colleagues assessed cognitive function in 183 patients approximately 7 years after KT. One third of these patients showed an impaired memory, about half impaired attention and mental processing speed and about 40% impaired executive function [6]. In another study Gelb and colleagues examined cognitive function in 42 patients approximately 5 years after KT and compared their results to 45 patients with chronic kidney disease and 49 healthy controls [7]. Both patient groups showed worse verbal memory and worse executive functioning skills than controls, while there was no significant difference between the two patient groups. Both studies indicate that cognitive dysfunction is present in KT patients in the long-term after transplantation. However, the underlying cause was not addressed. Data showing that cognitive function significantly improves within the first year after KT indicate that a secondary decline of cognitive function might occur in the long-term. Cognitive deterioration after transplantation was previously described in liver transplantation patients [9] and we recently showed cognitive impairment in a subset of patients in the long-term after liver transplantation who had been treated with low dose CNI after they had developed significant renal dysfunction with standard CNI therapy. We hypothesized that these patients might be hypersensitive against CNI toxicity [8].

Corresponding to the literature our results show that cognitive impairment is present in patients after KT compared to healthy controls. Especially visuospatial/constructional abilities seem to be affected in the long-term. The impairment of immediate memory in patients 1 year after KT might be a residue from dialysis related encephalopathy as patients on dialysis were shown to have especially memory and attention deficits [25]. The impairment of visuospatial/constructional ability in the long-term after transplantation was previously described in patients 10 years after liver transplantation who were on a reduced CNI dose due to CNI induced kidney injury [8]. The liver transplantation patients included in this study who were on a standard dose immunosuppressive therapy regimen showed a similar visuospatial/constructional ability as the KT patients, however, probably due to a higher variability of the test results no significant difference compared to healthy controls was found. Taken together, similar to patients after liver transplantation, also patients after KT show cognitive impairment in the long-term after transplantation. This might be related to long-term CNI therapy. All patient groups—KT and liver transplantation patients—showed worse visuo-constructive

abilities than controls, while language seemed preserved in all groups and attention seemed preserved especially in the KT patients. Thus deficits of visuo-constructional abilities might be related to tacrolimus therapy–a variable that is shared by all patient groups, while the deficit in alertness could be a residuum of hepatic encephalopathy.

Arterial hypertension and impaired kidney function might have an influence on cognitive function. However, it can be assumed that the impact of both on cognitive function in our study cohort is less important than that of the assumed vulnerability towards tacrolimus. For example, patients 5 years after KT showed only slight deficit in the visuospatial/constructional ability and no overall cognitive impairment despite having arterial hypertension and impaired kidney function just like the other KT groups. Furthermore, the patients after liver transplantation showed cognitive impairment although they had a normal kidney function and less often arterial hypertension than KT patients.

To address two of the possible tacrolimus associated pathomechanisms underlying cognitive dysfunction in our patients we performed MRI and 31P-MRS to assess cerebrovascular injury and cerebral mitochondrial energy metabolism. Structural brain alterations have been described in patients with significant chronic kidney disease with and without dialysis [26]'[27]. However, the effect of KT upon brain structure is not well described. Gupta and colleagues performed MRI and psychometric testing in eleven patients before and three months after KT [28]. They showed an improvement of cognitive function accompanied by an improvement of brain white matter integrity measured by diffusion tensor imaging in tracts associated with memory and executive function after KT. Zhang and colleagues measured white matter structural connectivity using diffusion tensor imaging in 21 patients before and 1 month after KT [29]. Although the MRI values of patients did not normalize, the diffusion tensor imaging results showed a significant improvement after KT. In both studies the authors concluded that structural brain alterations in chronic kidney disease patients might be at least partially reversible after KT.

Considering the assumed neurotoxicity of tacrolimus and the increased prevalence of cardiovascular risk factors in patients after KT [12], alterations of brain structure have to be expected. Accordingly, we found an increased extent of periventricular hyperintensities in patients about 1 and 5 years after KT compared to healthy controls. Surprisingly, KT patients about 10 years after transplantation did not significantly differ from healthy controls and none of the KT patient groups differed significantly from healthy patients concerning white matter hyperintensities or ventricular widths. Other groups discussed that structural brain alterations are at least partially reversible after KT [28, 29], however the lack of a difference between the patients 10 years after KT and controls in our study might well be due to the variability of these parameters and the limited number of subjects included into the study.

Another tacrolimus associated pathomechanism underlying cognitive dysfunction in patients after KT is the possible impairment of the cerebral energy metabolism. Illsinger and colleagues showed in vitro that clinically relevant tacrolimus concentrations impair the mitochondrial energy metabolism in human umbilical endothelial cells [14]. Furthermore, tacrolimus induced alterations of mitochondrial function was described in human cell lines [30]. Both studies indicate that tacrolimus impairs the mitochondrial metabolism. Based on these in vitro studies we hypothesized that tacrolimus associated impairment of mitochondrial function might be connected to long-term cognitive impairment in patients after transplantation. We found significantly reduced ATP concentrations in patients after KT compared to healthy controls. Furthermore, PME concentrations correlated significantly to cognitive function.

Because ATP is the main energy source for brain cells and represents mitochondrial activity [31] the reduced ATP concentrations in KT patients indicate altered mitochondrial function and might be associated to cognitive impairment. PME concentrations represent membrane

turnover of the brain [32] and consequently are connected to brain metabolism. In conclusion, our results indicate an impaired brain metabolism in KT patients. Interestingly, the liver transplantation patient control group which as well received tacrolimus for immunosuppression showed no alterations of the brain energy metabolism compared to controls. Thus, other factors besides tacrolimus must be involved. Kidney dysfunction might play a significant role in this respect. It might reduce the ability of the kidney to protect from CNI toxicity or it leads to an altered pattern of tacrolimus metabolites inducing neurotoxicity. Significant impairment of cognitive function was recently described in patients long-term after liver transplantation who had a history of kidney dysfunction and still decreased GFR [8]. The liver transplantation patients selected for this study had no kidney dysfunction at study inclusion (Table 1). This might explain why the liver transplantation patients in this study had no altered mitochondrial energy metabolism despite receiving tacrolimus therapy in similar doses gaining similar blood levels. Thus, the individual vulnerability towards tacrolimus associated toxicity needs to be considered as well. In conclusion, the tacrolimus mean trough level or tacrolimus dose alone are not a sufficient explanation for altered brain energy metabolism. In accordance, in our study neither was correlated to cognitive function or the cerebral energy metabolism.

Several limitations apply to our study. It has a limited transferability to other centres and unfortunately no data from before KT was available. The results of our study might be influenced by underlying diseases such as diabetes, other immunosuppressants, especially prednisolone, and most patients were taking several other drugs besides tacrolimus. Unfortunately, patients after KT who only received tacrolimus since transplantation as well as patients with a CNI-free or prednisolone-free maintenance therapy since transplantation were not available. The sample size limits statistical power and inhibits a statement on clinical impact. Furthermore, the different underlying diseases of patients after liver transplantation and KT limit comparability, but also allow the conclusion that cognitive impairment in the long-term after liver and kidney transplantation cannot exclusively be explained by CNI toxicity.

## Conclusions

In conclusion our results indicate that KT is associated with cognitive impairment and alterations of the brain energy metabolism in the long-term. The underlying pathomechanism seems to be complex and besides many other factors tacrolimus might be involved. Further investigation in a bigger study cohort is needed to clarify the role of tacrolimus and the impact of several variables such as concomitant disorders, extent of kidney dysfunction, sex and others. Furthermore, brain ATP in patients with chronic kidney disease and cognitive impairment before transplantation needs to be analysed in the future.

## Supporting information

**S1 Dataset. This dataset contains all data underlying our results.**
(XLSX)

**S1 Table. This table contains the details of the pulse sequences.** TE: echo time; TI: Inversion time; TR: repetition time.
(DOCX)

**S1 Fig. This figure shows the scatterplot between the RBANS Total scale and the tacrolimus total dose.** r = 0.62, p = 0.69; RBANS: Repeatable Battery for the Assessement of Neuropsychological Status.
(TIF)

**S2 Fig. This figure shows the scatterplot between the RBANS Total scale and the GFR.** r = -0.15, p = 0.35; RBANS: Repeatable Battery for the Assessement of Neuropsychological Status; GFR: glomerular filtration rate.
(TIF)

**S3 Fig. This figure shows the scatterplot between the RBANS Total scale and Bilirubin.** r = -0.76, p = 0.64; RBANS: Repeatable Battery for the Assessement of Neuropsychological Status.
(TIF)

**S4 Fig. This figure shows the scatterplot between the RBANS Total scale and ATP.** r = 0.31, p = 0.06; RBANS: Repeatable Battery for the Assessement of Neuropsychological Status; ATP: Adenosine triphosphate.
(TIF)

**S5 Fig. This figure shows the scatterplot between the RBANS Total scale and the occipital PVH.** r = 0.002, p = 0.99; RBANS: Repeatable Battery for the Assessement of Neuropsychological Status; PVH: periventricular hyperintensities.
(TIF)

## Acknowledgments

The authors would like to express their gratitude towards Professor Hans Messner and Professor David Gjertson for the continuous support in planning and conducting this study. Furthermore, we would like to thank Dr. Christopher Randolph for providing the German version of the Repeatable Battery for the Assessment of Neuropsychological Status (RBANS).

## Author Contributions

**Conceptualization:** Henning Pflugrad, Patrick Nösel, Xiaoqi Ding, Birte Schmitz, Hannelore Barg-Hock, Mario Schiffer, Karin Weissenborn.

**Data curation:** Henning Pflugrad, Patrick Nösel, Xiaoqi Ding, Birte Schmitz, Karin Weissenborn.

**Formal analysis:** Henning Pflugrad, Patrick Nösel, Xiaoqi Ding, Karin Weissenborn.

**Funding acquisition:** Henning Pflugrad, Xiaoqi Ding.

**Investigation:** Henning Pflugrad, Patrick Nösel, Mario Schiffer, Karin Weissenborn.

**Methodology:** Henning Pflugrad, Patrick Nösel, Xiaoqi Ding, Birte Schmitz, Mario Schiffer, Karin Weissenborn.

**Project administration:** Henning Pflugrad.

**Resources:** Xiaoqi Ding, Heinrich Lanfermann, Hannelore Barg-Hock, Jürgen Klempnauer, Mario Schiffer.

**Software:** Heinrich Lanfermann.

**Supervision:** Mario Schiffer, Karin Weissenborn.

**Validation:** Henning Pflugrad, Xiaoqi Ding, Mario Schiffer, Karin Weissenborn.

**Visualization:** Henning Pflugrad.

**Writing – original draft:** Henning Pflugrad.

**Writing – review & editing:** Patrick Nösel, Mario Schiffer, Karin Weissenborn.

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
