## [Decision Letter · Decision Letter 0]

3 Jan 2020

PONE-D-19-27390

Brain function and metabolism in patients with long-term tacrolimus therapy after kidney transplantation

PLOS ONE

Dear Dr. Pflugrad,

Thank you for submitting your manuscript to PLOS ONE. After careful consideration, we feel that it has merit but does not fully meet PLOS ONE’s publication criteria as it currently stands. Therefore, we invite you to submit a revised version of the manuscript that addresses the points raised during the review process.

I would like to first apologize again for the time that it has taken to render a decision on your manuscript. As I wrote to you, I had a very difficult time securing a second review. I appreciate your understanding in this. Between the two reviews, there are a number of issues to be addressed, but these are all relatively minor and will mostly help to clarify certain aspects that are somewhat unclear at the moment and tone done some statements. I encourage you to follow each of the suggestions/comments that have raised by both Reviewers, especially those that concern the presentation of the data.

Finally, please ensure that you state how data can be accessed. The data availability statement says "Yes - all data are fully available without restriction", but it is not clear how others can obtain access.

We would appreciate receiving your revised manuscript by Feb 17 2020 11:59PM. To enhance the reproducibility of your results, we recommend that if applicable you deposit your laboratory protocols in protocols.io, where a protocol can be assigned its own identifier (DOI) such that it can be cited independently in the future. For instructions see: http://journals.plos.org/plosone/s/submission-guidelines#loc-laboratory-protocols

We look forward to receiving your revised manuscript.

Kind regards,

Niels Bergsland

Academic Editor

PLOS ONE

Journal Requirements:

Reviewers' comments:

Reviewer's Responses to Questions

**Comments to the Author**

1. Is the manuscript technically sound, and do the data support the conclusions?

Reviewer #1: Yes

Reviewer #2: Yes

2. Has the statistical analysis been performed appropriately and rigorously? 

Reviewer #1: Yes

Reviewer #2: Yes

3. Have the authors made all data underlying the findings in their manuscript fully available?

Reviewer #1: No

Reviewer #2: Yes

4. Is the manuscript presented in an intelligible fashion and written in standard English?

Reviewer #1: Yes

Reviewer #2: Yes

5. Review Comments to the Author

Reviewer #1: The manuscript entitled "Brain function and metabolism in patients with long-term tacrolimus therapy after

kidney transplantation" describes the differences in cognitive function and brain ATP levels in kidney transplanted patients, liver tr4ansplanted patients, and healthy controls. The conclusions are, as expected, that kidney transplantation does not restore cognition to normal levels. This has been already mentioned in several reviews on this topic, as well as summarized in a recent review from a Europen group (see e.g. PMID:31071220 ).

However, I find that the manuscript contains important information that merit publication, specifically the use of a liver transplanted group as further control to dissect the effect of tacrolimus from kidney disease.

-the authors should greater strength this latter aspect in the title and the abstract

- the tables are difficult to read: please present the data in graphic form. I would encourage to present a graph with time in the horizontal axis (so that possible time-evolution is evident). (see below regarding the use of symbols)

- in figure 3 please use symbols that could better group the KT patients. e.g. use squares filled with at different gray levels to indicate KT 1, 5 and 10 years (rather than three different symbols). Why no error bars? why are the different cognitive measures connected by a line? this is misleading and makes confusion: please use for this type of graph bar plots.

- I do not agree with the conclusions: liver transplant seems to impinge even more on several cognitive aspects. Is there a dose-response relation between tacrolimus and cognition? please give a scatterplot of the dose of tacrolimus (and cumulative dose, if possible) vs cognitive function. Is it an effect linked to the eGFR (even in the case of liver transplant)? please give a scatterplot of eGFR vs cognitive functions. Is it a problem with liver function in kidney patients? please give a scatterplot of transaminases (or bilirubin, which is more likely to affect the brain) vs cognitive processes.

- ATP seems to be related to kidney transplants and not liver transplants. Therefore, it is not due to tacrolimus: please change the conclusions in the abstract and discuss the possibility that MCI-CKD is due to a loss of ATP into the brain due to CKD (see above reference for MCI-CKD). Please also give the relationship between ATP and cognitive functions (a regression and a scatterplot would be useful).

- I also find fascinating the trend effect of the PVH occipital, which is more present also in the liver transplanted patients. This is a very interesting point. You should demonstrate that these lesions are related to the cognitive measures (by regression, and possibly giving a scatterplot). In general, this finding is intriguing because the hyperintensities have been correlated usually to CKD rather than tacrolimus... is it possible that other drugs are actually causing them? maybe cortisone (which is widely used in CKD diseases)? or diabetes? what is the commonality between CKD, kidney transplant and liver transplant? maybe the data can offer some insight into this question.

Please reformulate the abstract once these points have been addressed.

Reviewer #2: The manuscript entitled “Brain function and metabolism in patients with long-term tacrolimus therapy after kidney transplantation” and submitted to PLOS ONE describes a study into the brain metabolism and cognitive function of kidney transplant recipients. Its cross-sectional study design analyzes patients 10, 5, and 1 years after transplantation under tacrolimus therapy, and compares these to both healthy controls and a group of patients after 10 years post liver transplantation. Methods included both neuropsychological tests, proton MRI and 31P (phosphorus) magnetic resonance spectroscopy. The results indicate that overall the kidney transplant recipients had cognitive impairment compared to healthy controls, but it did not follow that patients 10 years post-transplantation were more impaired than patients 5 and 1 years post.

It is curious that long-term tacrolimus therapy patients were not significantly more impaired than the 5 and 1 year patients, although as the authors state there may be other, more significant factors at work, and the sample size was not overly large. In general the study is well designed with the numerous statistical tests well described. A few keys area should be improved, however, in order to clarify the results and improve the text.

• The authors separate the patients and controls into 5 groups. Although they describe what each group represents (i.e. group 1 is kidney transplant patients 10 years post-transplantation), this gets confusing for the reader, and found myself making notes in order to keep the groups sorted correctly. It would be far easier for the reader if the authors altered their nomenclature for the groups, so that ‘KT10’ could represent kidney transplant patients 10 years post-transplantation. ‘KT5’, ‘KT1’, ‘LT’ and ‘HC’ would follow naturally, and the these should be carried into the various tables. This would be easier for the reader to follow.

• Figure 3 could also be improved as per the previous point. Currently, various symbols indicate which of the groups had significant (p < 0.05) deficits compared to controls. It would be far easier to just list the group names (i.e. LT5) as being significantly less than healthy controls, and state in the figure caption that all group means are being compared to controls.

• A 31P spectra would be interesting to the reader. Consider including an exemplary example as a figure, or at the minimum as supplementary material.

• Line 202-203: The turbo spin echo sequence appears to be a triple echo, with PD (13 ms), intermediate (71 ms) and T2-weighted (130 ms) echo times. It is important that the authors distinguish between this T2-weighted sequence, the T2 FLAIR sequence (see next point).

• Line 206: “an axial turbo inversion recovery magnetic sequence” Do the authors mean a T2-weighted fluid-attenuated inversion recovery (FLAIR) sequence? If so, please state that directly. You should also provide the parameters for this sequence (TR, TE, resolution, etc.) It may be worthwhile to create a new table for the pulse sequences, wherein all the relevant details (field of view, slice thickness, resolution, TR/TE, etc.) could be provided for all sequences. If the parameters were not kept consistent among the subjects (i.e. different FLAIR sequences were used) please state that directly so the reader can evaluate whether this had any effect upon the results.

• Line 208: “(WMH) were assess visually” On what the sequence? It’s almost certainly the FLAIR, but this should be stated directly. If other sequences (i.e. T2* weighted) were also used, this should be stated as well.

• Line 209-210: “ventricular width at the level...” Which sequence was used to measure these widths? The MPRAGE? FLAIR?

• Line 231: “exemplary T2 magnetic resonance image...” This image is almost certainly a T2-weighted FLAIR, not a T2-weighted sequence which was also described (Line 202-203). Please clarify the figure caption.

• Line 360: “Furthermore, KT patients showed enlarged periventricular hyperintensities...” Not sure what the authors mean by enlarged here. Do they mean an increased number of hyperintensities? The same number but increased volume?

• Line 418-419: “despite that they had arterial hypertension...” Consider changing to “despite having arterial hypertension...”

• Line 483-486: “Unfortunately, patients after KT who only received tacrolimus as well as a patient control group with a CNI-free or prednisolone-free immunosuppression since transplantation were not available.” This is not a complete sentence – please revise.

6. PLOS authors have the option to publish the peer review history of their article (what does this mean?). If published, this will include your full peer review and any attached files.

Reviewer #1: No

Reviewer #2: Yes: Paul Polak

---

## [Author Response · Author response to Decision Letter 0]

27 Jan 2020

Dear Professor Bergsland, dear Reviewers,

Thank you very much for the critical review of our manuscript. In the following we would like to respond to each of the comments. 

Academic editor:

Finally, please ensure that you state how data can be accessed. The data availability statement says "Yes - all data are fully available without restriction", but it is not clear how others can obtain access.

Answer: The S1 Dataset contains all data underlying our results.

Journal Requirements:

Answer: The manuscript and the names of the files were revised accordingly.

Comments to the Author

Have the authors made all data underlying the findings in their manuscript fully available?

Answer: All data underlying our results is now provided in the S1 Dataset.

Review Comments to the Author

Reviewer #1: The manuscript entitled "Brain function and metabolism in patients with long-term tacrolimus therapy after kidney transplantation" describes the differences in cognitive function and brain ATP levels in kidney transplanted patients, liver tr4ansplanted patients, and healthy controls. The conclusions are, as expected, that kidney transplantation does not restore cognition to normal levels. This has been already mentioned in several reviews on this topic, as well as summarized in a recent review from a Europen group (see e.g. PMID:31071220 ).

However, I find that the manuscript contains important information that merit publication, specifically the use of a liver transplanted group as further control to dissect the effect of tacrolimus from kidney disease.

-the authors should greater strength this latter aspect in the title and the Abstract

Answer: Thank you for this comment. The title was changed accordingly and the patient control group after liver transplantation was highlighted in the abstract.

- the tables are difficult to read: please present the data in graphic form. I would encourage to present a graph with time in the horizontal axis (so that possible time-evolution is evident). (see below regarding the use of symbols)

Answer: All data which is presented in the tables 2-4 is now additionally presented in graphic form (Fig 4-6). 

- in figure 3 please use symbols that could better group the KT patients. e.g. use squares filled with at different gray levels to indicate KT 1, 5 and 10 years (rather than three different symbols). Why no error bars? why are the different cognitive measures connected by a line? this is misleading and makes confusion: please use for this type of graph bar plots.

Answer: You are right. The figure was difficult to understand. Thus, we revised it completely. Now it is a bar plot graph with error bars (Fig 4).

- I do not agree with the conclusions: liver transplant seems to impinge even more on several cognitive aspects. Is there a dose-response relation between tacrolimus and cognition? please give a scatterplot of the dose of tacrolimus (and cumulative dose, if possible) vs cognitive function. Is it an effect linked to the eGFR (even in the case of liver transplant)? please give a scatterplot of eGFR vs cognitive functions. Is it a problem with liver function in kidney patients? please give a scatterplot of transaminases (or bilirubin, which is more likely to affect the brain) vs cognitive processes.

Answer: To assess whether tacrolimus has a dose-response impact on cognitive function and to assess whether the GFR or bilirubin have an impact on cognitive function we performed correlation analyses. The tests showed no significant relationship. All included patients after kidney transplantation had a normal liver function at study inclusion.

“The RBANS results of the KT patients did not correlate with the tacrolimus total dose (S1 Fig) or mean trough level, GFR (S2 Fig), Bilirubin (S3 Fig) or years on dialysis before KT.” (Lines 322-324)

We added the scatterplots of the correlation analysis between tacrolimus total dose, GFR as well as bilirubin and RBANS Total scale into the supplement (S1-S3 Figs). 

- ATP seems to be related to kidney transplants and not liver transplants. Therefore, it is not due to tacrolimus: please change the conclusions in the abstract and discuss the possibility that MCI-CKD is due to a loss of ATP into the brain due to CKD (see above reference for MCI-CKD). Please also give the relationship between ATP and cognitive functions (a regression and a scatterplot would be useful).

Answer: You are right. The reduction of ATP in the brain is only present in KT patients and not LT patients in this study. However, in a previous study we showed a reduced ATP in a subset of patients after LT who had developed kidney dysfunction due to calcineurin inhibitor therapy (Schmitz et al Aliment Pharmacol Ther. 2019). We hypothesized that some patients might be especially vulnerable towards CNI toxicity. The nine patients included in this study had no kidney dysfunction and thus do not belong to this group of patients. The results of our study show that patients after KT have cognitive dysfunction and reduced brain ATP. Considering former experimental data Tacrolimus may well be involved (see for example Illsinger et al, Annals of Transplantation 2011), however, the presence of renal dysfunction seems to play a significant role in these patients. To analyze the relationship between tacrolimus therapy, cognitive dysfunction, kidney function and other immunosuppressants further studies need to be done in the future with a bigger study cohort. The conclusion of the abstract was changed accordingly.

As you point out cognitive impairment in patients with chronic kidney disease might be due to lack of brain ATP. So far as we know, today no respective data are available. But it is of course an important question to follow up. We added a sentence to the conclusions to underline this aspect. 

“Furthermore, brain ATP in patients with chronic kidney disease and cognitive impairment before transplantation needs to be analysed in the future.” (Lines 561-563)

To display the relationship between ATP and cognitive function we performed a regression analysis and added a scatterplot into the supplement (S4 Fig). The statistical analysis was added to the methods section (Lines 258-263) and the results were added at the end of the results section (Lines 402-409).

- I also find fascinating the trend effect of the PVH occipital, which is more present also in the liver transplanted patients. This is a very interesting point. You should demonstrate that these lesions are related to the cognitive measures (by regression, and possibly giving a scatterplot). In general, this finding is intriguing because the hyperintensities have been correlated usually to CKD rather than tacrolimus... is it possible that other drugs are actually causing them? maybe cortisone (which is widely used in CKD diseases)? or diabetes? what is the commonality between CKD, kidney transplant and liver transplant? maybe the data can offer some insight into this question.

Answer: The occipital PVH were more present in all patients after transplantation compared to controls, however, the level of significance was only reached for the patients one year after KT. We extended the regression analysis by the occipital PVH and added a scatterplot showing the relationship between occipital PVH and the RBANS Total scale (S5 Fig). In regression analysis the occipital PVH had no significant effect on cognitive function (n=65, regression coefficient 2.34, p=0.24; 95% CI -1.64-6.32).

We agree that other drugs or underlying diseases such as diabetes might have an effect on our results. The number of patients included in this study makes it impossible to analyze effects of other drugs or underlying diseases on cognitive function and structural brain alterations. The study was designed as a pilot. We included these important aspects into the limitations of our study (Lines 543-553). 

The commonality between KT and LT patients in our study is the transplantation itself and the treatment with tacrolimus. The KT and LT patient groups, however, differed in regard to accompanying disorders such as presence of hypertension, renal dysfunction or diabetes. The impact of these factors as well as the impact of kidney dysfunction before transplantation should be analyzed in future studies. 

Please reformulate the abstract once these points have been addressed.

Answer: The abstract was revised to address the comments made above. 

Reviewer #2: The manuscript entitled “Brain function and metabolism in patients with long-term tacrolimus therapy after kidney transplantation” and submitted to PLOS ONE describes a study into the brain metabolism and cognitive function of kidney transplant recipients. Its cross-sectional study design analyzes patients 10, 5, and 1 years after transplantation under tacrolimus therapy, and compares these to both healthy controls and a group of patients after 10 years post liver transplantation. Methods included both neuropsychological tests, proton MRI and 31P (phosphorus) magnetic resonance spectroscopy. The results indicate that overall the kidney transplant recipients had cognitive impairment compared to healthy controls, but it did not follow that patients 10 years post-transplantation were more impaired than patients 5 and 1 years post.

It is curious that long-term tacrolimus therapy patients were not significantly more impaired than the 5 and 1 year patients, although as the authors state there may be other, more significant factors at work, and the sample size was not overly large. In general the study is well designed with the numerous statistical tests well described. A few keys area should be improved, however, in order to clarify the results and improve the text.

• The authors separate the patients and controls into 5 groups. Although they describe what each group represents (i.e. group 1 is kidney transplant patients 10 years post-transplantation), this gets confusing for the reader, and found myself making notes in order to keep the groups sorted correctly. It would be far easier for the reader if the authors altered their nomenclature for the groups, so that ‘KT10’ could represent kidney transplant patients 10 years post-transplantation. ‘KT5’, ‘KT1’, ‘LT’ and ‘HC’ would follow naturally, and the these should be carried into the various tables. This would be easier for the reader to follow.

Answer: Thank you for the recommendation. The nomenclature for the groups was changed accordingly.

• Figure 3 could also be improved as per the previous point. Currently, various symbols indicate which of the groups had significant (p < 0.05) deficits compared to controls. It would be far easier to just list the group names (i.e. LT5) as being significantly less than healthy controls, and state in the figure caption that all group means are being compared to controls.

Answer: Thank you for the advice. The Figure was revised accordingly (Fig 4). 

• A 31P spectra would be interesting to the reader. Consider including an exemplary example as a figure, or at the minimum as supplementary material.

Answer: Thank you for this suggestion. We included an exemplary 31P-MRS as Fig 3.

• Line 202-203: The turbo spin echo sequence appears to be a triple echo, with PD (13 ms), intermediate (71 ms) and T2-weighted (130 ms) echo times. It is important that the authors distinguish between this T2-weighted sequence, the T2 FLAIR sequence (see next point).

Answer: Thank you for the comment. The description of the MRI protocol was revised and is now described in more detail (Please see lines 199-239). 

• Line 206: “an axial turbo inversion recovery magnetic sequence” Do the authors mean a T2-weighted fluid-attenuated inversion recovery (FLAIR) sequence? If so, please state that directly. You should also provide the parameters for this sequence (TR, TE, resolution, etc.) It may be worthwhile to create a new table for the pulse sequences, wherein all the relevant details (field of view, slice thickness, resolution, TR/TE, etc.) could be provided for all sequences. If the parameters were not kept consistent among the subjects (i.e. different FLAIR sequences were used) please state that directly so the reader can evaluate whether this had any effect upon the results.

Answer: Thank you for this comment and please excuse the imprecision. The description of the MRI protocol was revised and now includes the asked details (Please see lines 203-220). The FLAIR sequence is now named directly and we provided all relevant details for the pulse sequences in S1 Table. The parameters were kept consistent among all subjects.

• Line 208: “(WMH) were assess visually” On what the sequence? It’s almost certainly the FLAIR, but this should be stated directly. If other sequences (i.e. T2* weighted) were also used, this should be stated as well.

Answer: Thank you for the comment and please excuse the imprecision. The WMH were assessed using the FLAIR and the GRE sequences. We added this information to the methods section. 

“Periventricular hyperintensities (PVH) and white matter hyperintensities (WMH) were assessed visually using the FLAIR and GRE sequences and semi quantitative by the Scheltens Scale” (Lines 215-217).

• Line 209-210: “ventricular width at the level...” Which sequence was used to measure these widths? The MPRAGE? FLAIR?

Answer: Thank you for the comment. The FLAIR sequence was used to measure the ventricular widths. 

“Furthermore, the ventricular width at the level of the caudate nucleus (VWCN) and semioval centre (VWSC) were measured in mm using the FLAIR sequence.” (Lines 217-219)

• Line 231: “exemplary T2 magnetic resonance image...” This image is almost certainly a T2-weighted FLAIR, not a T2-weighted sequence which was also described (Line 202-203). Please clarify the figure caption.

Answer: Please excuse the imprecision. A T2-weighted-fluid-attenuated inversion recovery sequence (FLAIR) was used. The figure caption was changed accordingly. 

 “This exemplary FLAIR image displays the assessed MRI values …” (Lines 241-245)

• Line 360: “Furthermore, KT patients showed enlarged periventricular hyperintensities...” Not sure what the authors mean by enlarged here. Do they mean an increased number of hyperintensities? The same number but increased volume?

Answer: Thank you for the comment and please excuse the ambiguity. The periventricular hyperintensities were assessed using the Scheltens Scale which is a semiquantative assessment tool. The extent of the periventricular hyperintensities was measured in mm and transformed into a score ranging from 0 to 2 (0=absent, 1= ≤5mm, 2= >5mm <10mm).

The sentence was clarified. “Furthermore, KT patients showed an increased extent of periventricular hyperintensities and reduced brain ATP concentrations compared to healthy controls.” (Lines 419-420)

• Line 418-419: “despite that they had arterial hypertension...” Consider changing to “despite having arterial hypertension...”

Answer: Thank you for the correction. The sentence was changed accordingly. (Lines 476-479)

• Line 483-486: “Unfortunately, patients after KT who only received tacrolimus as well as a patient control group with a CNI-free or prednisolone-free immunosuppression since transplantation were not available.” This is not a complete sentence – please revise.

Answer: Thank you for the comment. The sentence was revised.

“Unfortunately, patients after KT who only received tacrolimus since transplantation as well as patients with a CNI-free or prednisolone-free maintenance therapy since transplantation were not available.” (Lines 547-549)

We would like to express our thanks for the thorough review of our manuscript and lots of fruitful suggestions and comments. We hope that we have satisfactorily addressed all comments made by the reviewers and the academic editor and that you will now find our manuscript acceptable for publication in PLOS ONE.

---

## [Decision Letter · Decision Letter 1]

14 Feb 2020

Brain function and metabolism in patients with long-term tacrolimus therapy after kidney transplantation in comparison to patients after liver transplantation

PONE-D-19-27390R1

Dear Dr. Pflugrad,

We are pleased to inform you that your manuscript has been judged scientifically suitable for publication and will be formally accepted for publication once it complies with all outstanding technical requirements.

With kind regards,

Niels Bergsland

Academic Editor

PLOS ONE

Additional Editor Comments (optional):

Reviewers' comments:

Reviewer's Responses to Questions

**Comments to the Author**

1. If the authors have adequately addressed your comments raised in a previous round of review and you feel that this manuscript is now acceptable for publication, you may indicate that here to bypass the “Comments to the Author” section, enter your conflict of interest statement in the “Confidential to Editor” section, and submit your "Accept" recommendation.

Reviewer #1: All comments have been addressed

Reviewer #2: All comments have been addressed

2. Is the manuscript technically sound, and do the data support the conclusions?

Reviewer #1: Yes

Reviewer #2: Yes

3. Has the statistical analysis been performed appropriately and rigorously? 

Reviewer #1: Yes

Reviewer #2: Yes

4. Have the authors made all data underlying the findings in their manuscript fully available?

Reviewer #1: Yes

Reviewer #2: Yes

5. Is the manuscript presented in an intelligible fashion and written in standard English?

Reviewer #1: Yes

Reviewer #2: Yes

6. Review Comments to the Author

Reviewer #1: The revised version is now greatly improved; the authors have implemented all requested changes. The manuscript merits publication

Reviewer #2: (No Response)

7. PLOS authors have the option to publish the peer review history of their article (what does this mean?). If published, this will include your full peer review and any attached files.

Reviewer #1: No

Reviewer #2: Yes: Paul Polak

---

## [Editor Report · Acceptance letter]

26 Feb 2020

PONE-D-19-27390R1 

Brain function and metabolism in patients with long-term tacrolimus therapy after kidney transplantation in comparison to patients after liver transplantation 

Dear Dr. Pflugrad:

I am pleased to inform you that your manuscript has been deemed suitable for publication in PLOS ONE. Congratulations! Your manuscript is now with our production department. 

With kind regards,

on behalf of

Dr. Niels Bergsland 

Academic Editor

PLOS ONE